# Unsupervised Discovery of Interpretable Latent Manipulations in Language VAEs

## Abstract

Language generation models are attracting more and more attention due to their constantly increasing quality and remarkable generation results. State-of-the-art NLG models like BART/T5/GPT-3 do not have latent spaces, therefore there is no natural way to perform controlled generation. In contrast, less popular models with explicit latent spaces have the innate ability to manipulate text attributes by moving along latent directions. For images, properties of latent spaces are well-studied: there exist interpretable directions (e.g. zooming, aging, background removal) and they can even be found without supervision. This success is expected: latent space image models, especially GANs, achieve state-of-the-art generation results and hence have been the focus of the research community. For language, this is not the case: text GANs are hard to train because of non-differentiable discrete data generation, and language VAEs suffer from posterior collapse and fill the latent space poorly. This makes finding interpetable text controls challenging. In this work, we make the first step towards unsupervised discovery of interpretable directions in language latent spaces. For this, we turn to methods shown to work in the image domain. Surprisingly, we find that running PCA on VAE representations of training data consistently outperforms shifts along the coordinate and random directions. This approach is simple, data-adaptive, does not require training and discovers meaningful directions, e.g. sentence length, subject age, and verb tense. Our work lays foundations for two important areas: first, it allows to compare models in terms of latent space interpretability, and second, it provides a baseline for unsupervised latent controls discovery.

## 1 Introduction

Transformer-based models yield state-of-the-art results on a number of tasks, including representation learning (Devlin et al., 2019; Liu et al., 2019; Clark et al., 2020) and generation (Radford et al.; Raffel et al., 2019; Lewis et al., 2020). Notably, large language models have been reported to produce outputs nearly indistinguishable from human-written texts (Brown et al., 2020).

Although the predictions of autoregressive language models are fluent and coherent, it is not clear how to manipulate the model to get samples with desired properties. For example, make them shorter, more formal or more positive, or, alternatively, use the same model to rewrite human-written texts in a different tone. Current approaches often rely on external labels of target attributes and require modifications to the model. This involves retraining for new attributes or changing the decoding procedure, which is usually expensive.

In contrast, models with explicit latent spaces have the innate ability to manipulate text attributes by moving along latent directions. They, however, gained limited traction. One reason is that training a VAE on text data poses a number of optimization challenges, which have been tackled with a varying degree of success (He et al., 2019; Fu et al., 2019; Zhu et al., 2020). Additionally, language VAEs are mostly small LSTM-based models which goes against the current trend of using large pretrained Transformers. The first large-scale language VAE model is the recently introduced OPTIMUS (Li et al., 2020): it uses BERT as the encoder and GPT-2 as the decoder, and sets a new record on benchmark datasets.

Differently from texts, latent space models for images, especially GANs, achieve state-of-the-art generation results. Therefore, these models have been the focus of the research community, and the

properties of latent spaces are well-learned. For example, even early works on generative adversarial networks for images report that it is possible to have smooth interpolations between images in the latent space (Goodfellow et al., 2014). More recent studies show that the latent space directions corresponding to human-interpretable image transformations (from now on, "interpretable directions") can be discovered in an unsupervised way (Härkönen et al., 2020; Voynov & Babenko, 2020; Peebles et al., 2020).

In this paper, we show that for the language domain, much alike the well-studied visual domain, a sufficiently "good" latent space allows to manipulate sample attributes with relative ease. To avoid the known difficulties associated with training language GANs, we experiment with VAEs; more specifically, with the current state-of-the-art model OPTIMUS. We show that for this model, not only it is possible to produce meaningful and "smooth" interpolations between examples and to transfer specific properties via arithmetic operations in the latent space, but it is also possible to *discover the interpretable latent directions in an unsupervised manner*. We propose a method based on the PCA of latent representations of the texts in the training dataset. According to human evaluation, the proportion of interpretable directions among the ones found by our method is consistently larger than the proportion of interpretable directions among canonical co-ordinates or random directions in the latent space. The meaningful directions found by this method include, for example, subject age, subject gender, verb tense, and sentence length. Some of the directions, e.g. sentence length, are potentially useful: the ability to expand or shrink a text while preserving its content may be useful for tasks like summarization.

Note that the proposed method is simple and fast. The method is simple because it requires only the forward pass of the encoder, without backpropagating through decoding steps. This is very important for the language domain, where backpropagation through samples is significantly more difficult than for images. Namely, generation is non-differentiable, and previous attempts to overcome this issue relied on noisy or biased gradient estimates, which is less reliable than the standard MLE training. Instead, we do not rely on generated samples at all: we operate directly in the latent space. Additionally, since sampling directly from the prior does not yield diverse samples in case of OPTIMUS, we use the representations of the training data without running a decoding procedure - this maked the method fast.

To summarize, our contributions are as follows:

1. We propose the first method for unsupervised discovery of interpretable directions in latent spaces of language VAEs.

2. This method is simple and fast: it is based on PCA of latent representations for texts in the training dataset.

3. This method is effective: the proportion of interpretable directions among the ones found by our method is consistently larger than that of canonical co-ordinates or random directions in the latent space.

4. Our work lays foundations for two important areas: first, it allows to compare models in terms of latent space interpretability, and second, it provides a baseline for unsupervised latent controls discovery.

## 2 RELATED WORK

Finding interpretable directions in latent spaces of language VAEs is related to three lines of work. First, latent variable models for text and, more specifically, properties of latent spaces: for interpretable directions to exist, latent space has to be smooth (i.e. allow coherent interpolations). Then, since great part of the motivation for finding interpretable directions is manipulating generated texts, we discuss works on controllable text generation for different types of models, both VAE and standard autoregressive. Finally, we mention recent works trying to discover interpretable directions in image GANs.

### 2.1 LATENT VARIABLE MODELS FOR TEXT

Latent variable models encode information about text into a probability distribution. In addition to sampling new sentences from the prior distribution, they potentially allow to explicitly encode

specific properties of text, such as sentiment or style. Even early works on VAEs show that a latent space obtained with the VAE objective can result in coherent interpolations (Bowman et al., 2016).

While this is encouraging, training good VAEs with smooth and expressive latent spaces is challenging. Specifically, for interpretable directions to exist, we need a model which (i) does not ignore latent variable – to produce good samples, (ii) has continuous latent space – to allow controllable manipulation.

Ignoring latent variable is a known problem of VAEs. It arises because of the the KL vanishing problem: over the course of training, the KL divergence part of the loss may drop to 0, which indicates that the model ignores the latent variable. There exist many ways to alleviate this issue (Yang et al., 2017; Fang et al., 2019; Fu et al., 2019; Zhu et al., 2020); one of the simpler ways is adjusting the weight of the KL loss component according to a specific schedule.

Another problem is the latent vacancy problem: differently from images, not all regions of the latent space are occupied by the posterior distribution (Xu et al., 2020). In simple words, text latent spaces tend to have "holes" where the decoding network fails to generalize. As a result, when the latent codes are manipulated, the modified codes often land in these holes or vacant regions in the posterior latent space. If this happens, a model can not decode properly.

In light of the above, discovery of interpretable directions in text latent spaces is possible only with a strong model. Therefore, we use the current state-of-the-art model OPTIMUS (Li et al., 2020). It is a recent large-scale variational autoencoder which initializes the encoder with BERT (Devlin et al., 2019) and the decoder with GPT2 (Radford et al.). In addition to the model's high capacity, we use it because of the available checkpoints and reported results on latent space manipulation.

## 2.2 CONTROLLABLE GENERATION FOR TEXT DATA

**Latent variable models.** A natural way to achieve text generation with required attributes is using latent variable text generation models. The idea is that information about the attribute value is encoded in the latent code, and to obtain samples with the desired property one has to fix the corresponding component (direction) of the code.

For example, several works learn latent spaces with disentangled representations of content and style (Hu et al., 2017; Logeswaran et al., 2018; Lample et al., 2019; Yang et al., 2018; Shen et al., 2017; John et al., 2019). After that, to generate sentences in a specific style, the style vector is fixed. Depending on the approach, this style vector can either be estimated by encoding sentences with the desired attribute or be directly produced by specifying the structured latent code (e.g. one-hot encoding of an attribute).

Another line of research shows that it is possible to achieve attribute manipulation by moving in the latent space along specific vectors. These vectors, however, are found using data labelled with the attribute, i.e. with supervision. For example, Shen et al. (2020) change tense of a sentence by adding to its latent representation a "tense vector" computed as a difference of averaged representations of sentences with different tenses; Wang et al. (2019) use gradients of the attribute classifier. One of the first successful methods that learns a disentangled latent space is the work by Xu et al. (2020): they use basis vectors in the constrained latent space; however, this involves training a model with a structured latent space, which is rather complicated.

**Autoregressive models.** Controllable generation for standard autoregressive language models is usually achieved by either prepending an attribute to the input sequence as a prompt (Keskar et al., 2019), training an additional component of the model (Chan et al., 2020) or adjusting the decoding result with additional attribute-specific language models (Dathathri et al., 2020). A more thorough comparison of approaches to controlled text generation can be read in Prabhumoye et al. (2020).

Note that all these approaches require supervision and substantial changes to either training or generation procedures, whereas our approach is applicable to any variational autoencoder.

## 2.3 INTERPRETABLE DIRECTIONS MINING IN IMAGE GANs

To the best of our knowledge, there are only three works which discover interpretable latent directions in an unsupervised way, and all of them operate with GANs for images.

Two of them are not applicable to texts directly. In Voynov & Babenko (2020), the interpretable directions are trained. These directions are the ones which can be recognized by a separate reconstructor based on two samples, from the original and a shifted latent vector. Peebles et al. (2020) propose to learn disentangled attributes by minimizing the sum of squared off-diagonal terms of the generator Hessian matrix. Both approaches require backpropagation through sampling and therefore are not applicable directly for texts: unlike images, generated texts are not differentiable with respect to their latent representations.

The last approach, Härkönen et al. (2020), show that interpretable controls for image synthesis can be identified by finding principal components from layer outputs of the generator network for several samples from the prior. In our more challenging language domain, instead of sampling from the generator distribution, we take advantage of the availability of the encoder in VAEs and perform PCA on training data representations.

## 3 BACKGROUND

### 3.1 VARIATIONAL AUTOENCODERS FOR TEXT DATA

Variational autoencoders (Kingma & Welling, 2013) are a class of generative models. They consist of an encoder and a decoder. For a given text, the encoder outputs a latent code: a $d$-dimensional probability distribution (usually a gaussian with diagonal covariance matrix). The decoder receives a vector $z$ sampled from some distribution (in training, from the encoder output, in inference, from the prior) and generates an output text.

VAEs are trained to minimize the sum of the reconstruction loss (for language, usually it is the cross-entropy loss) and the KL divergence loss, which forces the encoder outputs to stay close to the prior distribution. To avoid the KL vanishing problem (see section 2.1), the KL term of the loss is weighted with a $\beta$, which is annealed according to a predefined schedule (Bowman et al., 2016). Formally, the loss function is as follows:

$$\arg\min_{\phi,\theta} \mathcal{L}_\beta = \arg\min_{\phi,\theta} \left( \mathcal{L}_{rec} + \beta\mathcal{L}_{KL} \right) \tag{1}$$

For more details on the specifics of loss and training, see e.g. (Li et al., 2020).

### 3.2 OPTIMUS: THE FIRST LARGE-SCALE TEXT VAE

The current state-of-the-art variational autoencoder is OPTIMUS (Li et al., 2020). This is the first large-scale VAE: it initializes encoder and decoder networks with pretrained weights of BERT (Devlin et al., 2019) and GPT-2 (Radford et al.) respectively. There are two strategies for including the latent vector in the generative process of the Transformer decoder: memory and embedding. In the memory strategy, latent vector is concatenated with the prefix tokens; in the embedding strategy, this latent vector is added to every token representation in the decoder.

We use OPTIMUS because for our study, we need a strong model with a good latent space. For a detailed explanation, see section 2.1.

## 4 OUR METHOD

Each method for discovery of interpretable directions is inevitably paired with its specific underlying definition of the vague notion of "interpretability": the former defines the latter, and vice versa. For example, in Voynov & Babenko (2020), interpretable directions are the ones that are easy to distinguish from each other by observing results of manipulations; in Peebles et al. (2020) — the ones that change the output image independently from each other.

To start, we formulate the desired properties we believe interpretable directions should have. In our setting, interpretable directions are the ones that:

- are *orthogonal:* this is a weaker form of the standard attribute disentanglement requirement, which states that changing one attribute of the sentence should not affect others;

- *maximize variance,* i.e. they "explain" the data distribution well, so that its most distinctive features are exposed as controllable attributes. This requirement might be formalized as follows: given the $d$-dimensional latent representations of $n$ samples $X \in \mathbb{R}_{n \times d}$, we are interested in finding a set of vectors $w_i$ that maximize the sum of $||Xw_i||^2$.

## 4.1 PCA FOR ENCODED TRAINING DATA

Luckily, a common statistical technique Principal Component Analysis (PCA) satisfies both of these requirements by construction. This technique obtains principal components (vectors with highest variance) as eigenvectors of the covariance matrix $X^T X$ corresponding to the largest eigenvalues. Although PCA is mostly used as a dimensionality reduction method, we use it to obtain a linear transformation that corresponds to a set of disentangled controllable text attributes.

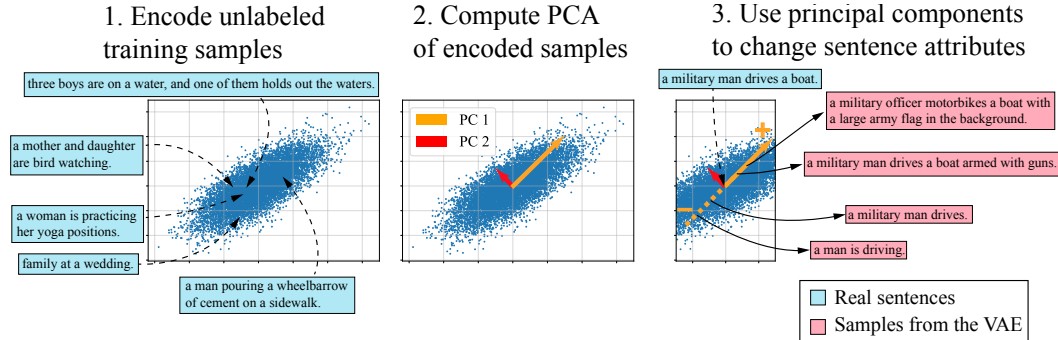

Figure 1: The illustration of our method; PC stands for "Principal Component".

The resulting method is a straightforward procedure (see Figure 1):

1. Encode examples from the training dataset (since only encoder is used, no generation is required). For each example, we will have a probability distribution.
2. Take the expectation vector of each distribution, stack them to form a matrix.
3. Compute the principal components of the resulting matrix.
4. Take top-N principal components with the highest explained variance.

Note that our method is very simple: it does not involve neither training any additional models nor generation from the model (which is slow for texts). Despite its simplicity, this approach discovers meaningful and often non-trivial latent manipulations (section 5.2) and the proportion of interpretable directions is greater compared to several baselines (section 5.3).

## 4.2 NOTES ON DESIGN CHOICES

In this part, we explain some of our design choices in light of related works on interpretable directions mining for image GANs (Voynov & Babenko, 2020; Peebles et al., 2020; Härkönen et al., 2020) and highlight why the fist two of these three methods are not directly applicable for texts.

**We do not sample from the prior.** An important distinction of our approach from prior work on latent discovery in image GANs is that we do not use sampling from the prior distribution of the trained model. This has several reasons. First, while image generators are convolutional and the entire image is generated in one pass, text decoders produce sentences token by token, which significantly slows down training. Second, we observed that sampling from the prior distribution of OPTIMUS does not result in high-quality samples. For example, when latent vectors are sampled from $\mathcal{N}(0, I_d)$, the resulting texts are highly repetitive. Namely, on average over $40\%$ of tokens in a generated text have the same token type. To achieve higher generation quality, we could estimate the variance $\sigma(z)$ from training data and sample latent vectors from $\mathcal{N}(0, \sigma(z))$ instead of $\mathcal{N}(0, I_d)$. However, this requires mapping all training sentences to the latent space; at that point, one already has latent vectors encoding natural texts, so further sampling is redundant.

**Two of the methods are not directly applicable for texts.** While our method is similar to the one proposed in Härkönen et al. (2020) for images, there exist two other methods that show promising results. Here, we would like to stress that approaches by Voynov & Babenko (2020) and Peebles et al. (2020) for the visual domain are not directly applicable to language generation: they require backpropagation through discrete sampling. This problem has previously been studied in the context of language GAN training, and there is an evidence that the community has not yet proposed a solution that would consistently outperform maximum likelihood training (Caccia et al., 2020).

**Other matrix decompositions.** There exist numerous ways to decompose the latent representation matrix into several orthogonal components. However, our preliminary evaluation of Independent Component Analysis (Hyvärinen & Oja, 2000), which is one of the most widely used alternatives to PCA, did not yield substantially better results in terms of direction interpretability.

## 5 EXPERIMENTS

In this part, we first show qualitative results: categories of the interpretable directions found by our method along with some examples. Next, we show that the success of our method can not be attributed solely to the properties of the latent space. Namely, the proportion of interpretable directions among the ones found by our method is larger than that of the baselines. Since, as we will see, types of the revealed directions can be quite diverse, developing a metric for automatic evaluation of interpretability is challenging. Therefore, we resort to human evaluation.

### 5.1 SETUP

**Models.** We evaluate three publicly available OPTIMUS checkpoints: one of the model trained on a dump of Wikipedia with $\beta = 0.5$ and latent vector size 32, and two of the models finetuned on the SNLI dataset (Bowman et al., 2015) with $\beta = 0$ and $\beta = 1$ and latent vector size 768.[1]

In preliminary experiments, we also examined other models from prior work. As expected, simple attributes are discoverable even in lower capacity models, but the overall reconstruction and generation quality is worse.

**Finding directions.** We compute the PCA for latent representations on a subsample of 50000 training examples for each model; larger subsample sizes did not improve quality of the attributes.

**Baselines.** Since mining of interpretable latent manipulations for texts is a new task, there are no establied methods. Therefore, we implement two simple but reasonable baselines.

- *Random directions:* vectors sampled from the standard normal distribution and normalized to unit length. This is the easiest way to find meaningful directions in the latent space.
- *Coordinate directions:* coordinate vectors in the latent space. To choose the "best" directions, we pick coordinates with the highest variance over the training dataset representations. This goes in line with the motivation of PCA: out of all possible coordinate directions, these are responsible for most of the variance in data. [2]

Note that the coordinate baseline can be potentially quite strong. The standard Gaussian distribution, which is the prior for the OPTIMUS model, has independent components. Since OPTIMUS is trained with KL divergence between the posterior and the prior distributions, canonical coordinates in its latent space are likely to be disentangled.

**Evaluating directions.** To evaluate the directions, we sample sentences from the corresponding training set and apply the latent shift to the encoder output. Namely, we increase or decrease the

---

[1]Ideally, we would examine the properties of a model with a high-dimensional latent space trained on a general corpus. Unfortunately, this was not possible: training such a model from scratch is prohibitively expensive, and OPTIMUS checkpoints from this kind of models are not publicly available.

[2]Additionally, in preliminary experiments we also looked at top coordinates with the lowest variance, as well as randomly chosen ones. Coordinates with the highest variance tend to be more reasonable and interpretable.

attribute presence by multiplying the shift vector by a scalar constant and adding it to the mean vector of the distribution. The value of the scalar varies from $-m$ to $m$; in each case, we take 5 points, $\{-m, -0.5m, 0, 0.5m, m\}$, so that we observe the unchanged sentence and its 4 modifications with a varying direction and degree of change. For the PCA directions, $m = 5$; for the baselines, $m = 10$. These values of $m$ are found empirically: lower $m$ result in mostly no changes in model output, higher lead to degenenerate sequences with repeated tokens. For generation, we use nucleus sampling (Holtzman et al., 2020) with $p = 0.9$; in preliminary experiments, we also found the results do not change when using greedy decoding.

## 5.2 QUALITATIVE RESULTS

First, we observe that almost all variance in latent representations of the SNLI is covered by a small fraction of principal components (see Appendix A). This hints at the presence of directions that capture the most important features of these texts.

Next, we look at how the shifts along the top components influence generated texts. Figure 2 shows examples of directions discovered by our method for the model trained on the SNLI dataset. The discovered directions can be roughly categorized into four types:

1. *Basic sentence attributes*, such as length: varying the length attribute produces expanded or contracted versions of the original sentence with minor content alterations. This might be useful for sentence summarization or the reverse task of enriching a text with more details.

2. *Word-level attributes*: similarly to image models, OPTIMUS has directions that can affect the age of the subject, transform nouns from singular to plural, etc. One of these directions transforms nouns from gender-specific to gender-neutral (e.g. "a man" / "a woman" changes to "a person"); this can also find practical applications given the known issue of unintended biases in language models (Dinan et al., 2020; Florez, 2019).

3. *Insertion of particular words*: other than changing the existing words in the sentence, some directions force the model to add specific tokens. They range from content words, such as nouns, to function words, such as prepositions.

4. *Enforcing specific structure*: a few directions change the input sentence to adhere to a certain syntactic structure. In case of SNLI, this may be explained by the declarative nature of most sentences in the dataset with several frequent verb phrases.

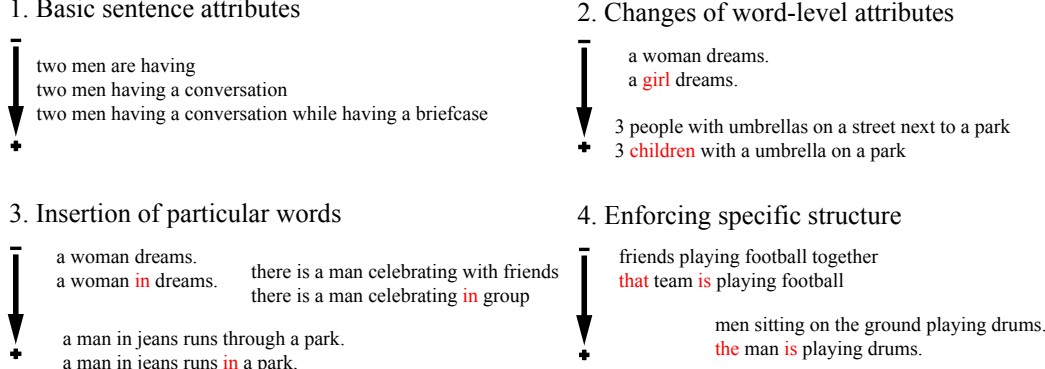

Figure 2: Examples of the directions discovered by our method for the SNLI model.

## 5.3 HUMAN EVALUATION

### 5.3.1 PROCEDURE

We evaluate the three available checkpoints of OPTIMUS and the three methods (PCA with two baselines) described above. For each model, we sample 20 sentences from the corresponding training dataset. Then, for each method we take 20 "best" directions and apply them to the gathered

sentences. Note that some directions are applicable only to a fraction of sentences (e.g., gender-specific attributes require a human subject). If after applying direction shifts to a sentence at least three of the 5 versions are the same, we do not use this sentence for this direction. To form an annotation task, we randomly choose 5 sentences from the filtered results. For a single combination of model, method, and direction, we generate 5 such tasks.

The evaluation protocol for one annotation task is as follows:

1. The annotator sees 5 sentences along with their modifications. They have to answer whether the direction corresponding to that example is interpretable.[3]

2. If the direction is interpretable, the annotator has to specify its category: either choose one the described in section 5.2 or enter a description manually.

3. If the direction is partially interpretable (e.g. content preservation is lacking or the results are "visible" only in part of the examples), the annotator has to indicate this.

Our annotators are 12 people with the background in machine learning. On average, we give each participant about 75 annotation tasks. For a single combination of model, method, and direction, we aggregate the results with the majority vote.

| Method | SNLI, $\beta = 0$ | SNLI, $\beta = 1$ | Wiki |
|---|---|---|---|
| Random | 0.2 | 0.3 | 0 |
| Coordinate | 0.2 | 0.15 | 0 |
| PCA | **0.6** | **0.4** | **0.05** |

Table 1: Proportion of interpretable directions found by each method.

| Method | Basic | Word Insertion | Structure |
|---|---|---|---|
| Random | 0.41 | 0.35 | 0.09 | 0.12 |
| Coordinate | 0.63 | 0.23 | 0.06 | 0.06 |
| PCA | 0.25 | 0.6 | 0.06 | 0.06 |

Table 2: Proportion of direction categories found by each method for the SNLI model.

### 5.3.2 RESULTS AND DISCUSSION

The results are shown in Tables 1 and 2; more detailed statistics are given in Appendix C. Our method outperforms the baselines for all three models, being the only one that achieves more than 50% of output direction interpretability. Apparently, the Wikipedia model exhibits too low reconstruction quality to be interpretable; this is most likely because its latent space dimension is 32, which is very small.

Interestingly, for the SNLI model with $\beta = 1$, coordinate directions perform worse than random direction or even coordinate directions for the model with $\beta = 0$. Let us recall that $\beta$ affects the regularization strength: higher $\beta$ forces the latent distribution to be closer to the prior, in this case, an isotropic Gaussian. This has two implications. First, the model with the stronger regularization ($\beta = 1$) has weaker representation capabilities. Second, differences in variance of the output distribution become less pronounced: as such, the fraction of explained variance becomes a non-informative coordinate selection criterion.

If we now look at the specific categories of the directions discovered by different methods (Table 2), we will see an interesting pattern: the baseline methods discover mainly basic text properties (e.g. length), but our method identifies mostly changes of word-level attributes (subject gender, age, number, etc.) Note that these properties are more useful for the SNLI task than basic sentence attributes. This suggests that the directions discovered by our method are more reasonable for a given model.

## 6 CONCLUSION

We propose the first method for unsupervised discovery of interpretable attribute manipulations in text variational autoencoders. This method involves computing the principal components of training data representations. It is very simple, fast, and outperforms the baselines by a large margin. Future work may investigate the locality of directions in text: many modifications (e.g. age) are applicable to only a fraction of sentences, and this might influence the results in Table 1.

---

[3]The instruction shown to the annotators can be seen in Appendix B.

We believe this work will encourage further research in two important areas. First, our approach may serve as a baseline for the new task of unsupervised discovery of interpretable latent controls in generative models. Second, it can be used to compare models in terms of latent space interpretability.

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

## A  VARIANCE EXPLAINED BY PRINCIPAL COMPONENTS

To verify that a small number of directions found by our method *can* explain the large proportion of variance in SNLI, we plot the cumulative explained variance ratio for each coordinate, ordered by explained variance in decreasing order. It can be seen that 70 latent directions are responsible for most of the variance in the training dataset: given good reconstruction capabilities of OPTIMUS, this means that almost all sentences of SNLI can be described in terms of these 70 orthogonal vectors.

## B  HUMAN EVALUATION INSTRUCTION

### B.1  TASK DESCRIPTION

In each assignment, you will see 5 sentences with a certain textual characteristic (or attribute) changed by the model. There are 5 degrees of change of different intensity on the scale of "–" to "++", the unchanged sentence is given in the middle.

1. You need to answer whether it is possible to interpret the changing attribute from given examples. If the atribute is partially interpretable (for instance, because the nature of changes is the same in only 2 of 5 sentences), you should also state this.

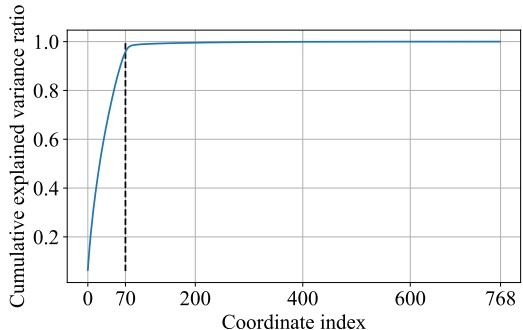

Figure 3: Cumulative explained variance with increasing the number of components.

2. If you can interpret the attribute (even partially), you need to assign it to one of the attribute categories or specify your own. Do not hesitate to give your own name to the attribute in the corresponding field: the model might change sentence properties that are not given in the list or do not fit the provided categories[4].

3. If attribute interpretation was challenging or only partial, choose the option "Partially interpretable" and give the exact reason: whether this is because of several properties changing at once, one property being interpretable in only 1-2 sentences out of 5, or too drastic content changes. You can also explain partial interpretability using the text field.

4. If the attribute is not interpretable because you cannot unambiguously formulate its essence so that it would fit all sentences, you should choose the option "No". If changing this attribute leads to sentences becoming meaningless character sequences or repetitions of the same words, you should choose the option "Incorrect text".

## B.2 EXAMPLES WITH DIFFERENT INTERPRETABILITY

**Example 1 (correct changes in the attribute):**

```
--   there is a man celebrating
-    there is a man celebrating
     there is a man celebrating with friends
+    there is a man celebrating with friends and celebrating in
front of a red building
++   there is a man celebrating with friends celebrating with a red
flag about to celebrate with other peoples in <unk>

--   a man is walking.
-    a man in jeans runs.
     a man in jeans runs through a park.
+    a man in jeans runs through a park through jean shorts.
++   a man in jeans jeans runs through a park in a gray t-shirt
through woods with runners past.
```

In this case, for both sentences changing the attribute leads only to changes in the sentence length, so the appropriate answer is "Yes".

**Example 2 (content not preserved):**

```
--   In the aftermath of the war, the local population of the
area, including the local inhabitants of the area, who had been
evacuated from the area by the Germans.
```

---

[4]Despite this sentence, the option of custom attribute type was mostly unused by all annotators.

```
-      In the aftermath of the war, the "Carnivores" were sent to
the "Carnivores" camp in the "Carnivores" area, where they were
killed.
       The remaining members of the "Cavaliers" were killed in the
battle, and the remaining members of the "Cavaliers" were killed
in the battle.
+    The remaining were:
++

--    In the early 1990s, after a series of interviews with the
author of "The Adventures of Dr. John D. MacDonald", who had
been diagnosed with Parkinson's disease, and his wife, Dr. Louise
MacDonald.
-      In "The Adventures of Doctor Who", he was introduced as a
young man who was sent to the Earth by the Doctor.
       She was played by John Hurt in the first film, "The Man Who
Wasn't Born".
+    She is played by John C. Reilly.
++

--    In addition to her high school education, she received a
$1,000 grant from the state of New York to provide a $1,000 grant
to the New York State Department of Education
-      In addition to her high school diploma, she received a
bachelor's degree in business administration from the University
of Michigan, and a master's degree in business administration from
the University of Michigan
       She earned a bachelor's degree in business administration,
and a master's degree in business administration.
+    5.
++
```

In this case the sentence length also changes, but it also leads to a drastic change in content (it is not possible to say that all sentences were obtained from the one in the middle by adding or removing words). In such cases, the expected answer is "Partially" and the reason is "Too drastic content changes".

**Example 3 (several attributes are changed at a time):**

```
--    a couple are embracing.
-      two people are embracing.
       two people are embracing each other.
+    two people are embracing each other across the surface.
++ two humans glide across each other as each other are embraced
by the other.

--    a couple with a umbrella
-      2 people with a umbrella on a beach
       3 people with umbrellas on a street next to a park
+    3 people with umbrellas on a street next to some pedestrians
outside a park
++ 5 pedestrians all on green spots along side a curb with a
street in front of some pedestrians in an area of nyc

--    two men and is a birthday.
-      two men and three children are at the beach.
       two men and three children are at the beach.
+    two men and three children are at the beach three.
++ three men and three children all sit at the side of the road in
green and yellow.
```

In this case, the meaning is mostly preserved, but there are two changes at the same time: sentence length and the number of subjects. Because of this, you should choose "Partially interpretable — More than one attribute changes".

**Example 4 (non-interpretable changes):**

```
--    three girls watch a ballet dance.
-     four women watch a ballet practice from their chair.
       four women watch a ballet practice from their chairs.
+     four women watch a ballet from their practice chairs.
++  women watch a ballet during their breaks at the computers.
```

```
--    young boy using a vacuum cleaner.
-     young boy using a vacuum cleaner on a rug.
       young boy using a vacuum cleaner on a rug.
+     young boy using a vacuum cleaner on a rug.
++  young boy using a vacuum on a dry rug.
```

```
--    a boy is jumping from a plank.
-     a boy jumps from a plank high.
       a boy jumps from a plank up high.
+     a boy jumps from a plank up high.
++  a boy jumps from a plank up over the water.
```

Here (at least, for the task authors) it is too hard to briefly explain the kind of changes, hence the answer should be "No".

**Example 5 (incorrect text):**

```
--    " " " " " " " " " " " " " " " " " " " " " " " " " " " " " " " "
" " " " " " " " " " " " " " " " " " " " " " " " " " " " " " " " " "
" " " " " " " " " " " " " " " " " " " " " " " " " " " " " " " " " "
" " " " " " " " " " " " " " " " " " " " " " " " " " " " " "
-     "The Merry Christmas" was signed by the London Company and
sold to the Royal Opera Company in 1885 for £1,000, and was the
first of the "Merry Christmas" to be sold to the Royal Opera
Company.
       The casualties were estimated at 1,000,000 men, and the
British were estimated at 1,000,000.
+    There are no casualties.
++
```

In this case significant changes to the attribute ("–" and "++") lead to the sentence no longer being readable. You may choose "The text is incorrect" if it is impossible to understand the interpretability of attribute in this case.

### B.3    ATTRIBUTE TYPES

Each attribute may be connected with simple sentence features (e.g length), presence of certain words in the sentence or even with sentence sentiment. You can see examples of different attributes below:

**Length:**

```
--    3 people are outside
-     3 people with a umbrella on street
       3 people with umbrellas on a street next to a park
+    3 umbrellas with a person on top of a street for umbrellas
outside a park
```

```
++ 3 umbrellas with umbrellas on a street corner with a lot of
people next to a green parka with a street sign on it in front of
it for an area of a city in an asian city
```

**Singular/plural number:**

```
--   a worker standing on a high scaffold.
-     a worker standing on a high scaffold.
      a worker standing on a high scaffold.
+    a worker standing on high scaffolding.
++ several workers standing high on the scaffold.
```

**Changing the order of words in a sentence (also, the number is changed):**

```
--    the are standing and reading a newspaper.
-     the man is standing and reading a newspaper.
      a man is standing and reading a newspaper.
+    a man is standing and reading in a newspaper.
++ a man is standing and running in a newspaper.
```

```
--     the naked men are holding the drum.
-      the naked man is playing drums.
       the naked man is playing drums.
+     a naked man is playing drums.
++  a naked man is playing in wate
```

**Adding the word "in":**

```
--     a man runs through jeans.
-      a man in jeans runs through a park.
       a man in jeans runs through a park.
+     a man in jeans runs through a park.
++  a man in jeans runs in a park.
```

## C  ADDITIONAL TABLES FROM THE HUMAN STUDY

Here, we provide full data on the results of the interpretability evaluation without aggregating results over different interpretability categories (Table 3) or different models for direction types (Table 4).

## D  EXAMPLES OF FOUND DIRECTIONS

Below we give additional examples of latent directions discovered by our method on the SNLI dataset ($\beta = 0$):

1. *Basic sentence attributes:*

```
--     children are playing a game.
-      children are playing a game.
       children are playing a game together.
+      children are playing a game together as a ball game.
++     children playing a game of board war together are
playing a game of snowball together after a birthday.
```

```
--     a man is looking down.
-      a fireman is looking down.
       a fireman is looking down at the ground.
+      a fireman looking down at the ground is still raining
down.
++     a firetruck driver looking down at the ground about to
fall down from the ground is still burning up smoke.
```

| Method | Model | Interpretable? | Proportion |
|---|---|---|---|
| Coordinate | snli_beta0 | no | 0.80 |
| | | yes | 0.15 |
| | | partial | 0.05 |
| | snli_beta1 | no | 0.85 |
| | | partial | 0.10 |
| | | yes | 0.05 |
| | wiki | not_coherent | 0.90 |
| | | no | 0.10 |
| PCA | snli_beta0 | partial | 0.45 |
| | | no | 0.40 |
| | | yes | 0.15 |
| | snli_beta1 | no | 0.60 |
| | | yes | 0.25 |
| | | partial | 0.15 |
| | wiki | no | 0.80 |
| | | not_coherent | 0.15 |
| | | partial | 0.05 |
| Random | snli_beta0 | no | 0.80 |
| | | partial | 0.15 |
| | | yes | 0.05 |
| | snli_beta1 | no | 0.70 |
| | | partial | 0.20 |
| | | yes | 0.10 |
| | wiki | not_coherent | 0.90 |
| | | no | 0.10 |

Table 3: Evaluation data for all interpretability categories

2. *Word-level attributes:*

Singular/plural noun:

```
--    family at a a wedding.
-     family at a wedding.
      family at a wedding.
+     family at a wedding.
++    families at wedding.

--    a a person performing a a guitar for a little crowd.
-     a band performing for a little crowd.
      a band performing for a little crowd.
+     the band performing for some big crowd.
++    the bands performing just for friends.

--    a child is playing a game a ball in a room.
-     children are playing a game together.
      children are playing a game together.
+     children are playing a game together.
++    children are playing game together.
```

Age:

```
--    the man is is important.
-     the man is athletic.
      the man is athletic.
+     the man is athletic.
++    the athletic boy.

--    a woman is sitting at her positions to practice their
positions.
```

| Method | Model | Direction type | Proportion |
|---|---|---|---|
| Coordinate | snli_beta0 | basic_properties | 0.86 |
| | | changes_object_properties | 0.07 |
| | | insert_word | 0.07 |
| | snli_beta1 | basic_properties | 0.44 |
| | | changes_object_properties | 0.38 |
| | | specific_structure | 0.12 |
| | | insert_word | 0.06 |
| PCA | snli_beta0 | changes_object_properties | 0.68 |
| | | basic_properties | 0.16 |
| | | specific_structure | 0.11 |
| | | insert_word | 0.05 |
| | snli_beta1 | changes_object_properties | 0.47 |
| | | basic_properties | 0.37 |
| | | insert_word | 0.1 |
| | | specific_structure | 0.05 |
| | wiki | changes_object_properties | 0.80 |
| | | basic_properties | 0.20 |
| Random | snli_beta0 | changes_object_properties | 0.53 |
| | | basic_properties | 0.29 |
| | | specific_structure | 0.12 |
| | | insert_word | 0.06 |
| | snli_beta1 | basic_properties | 0.58 |
| | | changes_object_properties | 0.17 |
| | | insert_word | 0.17 |
| | | specific_structure | 0.08 |
| | wiki | basic_properties | 0.50 |
| | | specific_structure | 0.50 |

Table 4: Evaluation data for all direction types, grouped by model and method

```
--      a woman is practicing her positions at an yoga center.
        a woman is practicing her yoga positions.
+       a female is practicing her yoga positions.
++      a yoga girl practicing her knees.

--      the woman woman is patting volleyball.
-       the women are playing volleyball in the park.
        the women are playing volleyball in the park.
+       the women are playing volleyball in the park.
++      the girls are playing volleyball in the park.
```
Gender-neutral/gender-specific:
```
--      children are playing a game together.
-       children are playing a game together.
        children are playing a game together.
+       children are playing a game together.
++      boys are play a game together.

--      a person is practicing her standing muscles.
-       a woman is practicing her positions yoga.
        a woman is practicing her yoga positions.
+       a woman is practicing her yoga positions.
++      two women practices her yoga positions.
```
3. *Insertion of particular words:*
   "In":
```
--      children are playing a game to celebrate.
-       children are playing a game together.
        children are playing a game together.
```

```
+     children are playing a game together.
++    two children in a game are sitting together.

--    the women are practicing volleyball playing volleyball.
-     the women are playing volleyball during the beach.
      the women are playing volleyball in the park.
+     the women are playing volleyball in the park.
++    the two women in volleyball are sitting in a park.
```
"Man":
```
--    children are playing a game in full.
-     children are playing a game together.
      children are playing a game together.
+     two children are playing a game together.
++    two men are playing a game together.

--    a person pouring a cup of concrete sidewalk in a window
display.
-     a person pouring a wheelbarrow of cement on a sidewalk.
      a man pouring a wheelbarrow of cement on a sidewalk.
+     a man pulling a wheelbarrow of cement on his bicycle.
++    a man pulling a wheelbarrow of two men and a hammer on
the rail.

--    the person is athletic.
-     the man is athletic.
      the man is athletic.
+     the man is athletic.
++    the man and his man are athletic.
```
4. *Enforcing specific structure:*
   "A is B":
```
--    children playing game together outside a house.
-     children are playing a game together.
      children are playing a game together.
+     children are playing a game together.
++    the child is a game having a winning.

--    women playing volleyball in the park.
-     several women are playing volleyball in the park.
      the women are playing volleyball in the park.
+     the women are playing volleyball in the park.
++    the woman is the volleyball player.
```
   "A is B in C"
```
--    the women are playing the volleyball bar.
-     the women are playing volleyball in the park.
      the women are playing volleyball in the park.
+     the woman are playing volleyball in the park.
++    a woman is playing volleyball in city.

--    the fireman are looking down the table.
-     the fireman is looking down at the ground.
      a fireman is looking down at the ground.
+     a fireman is looking down at ground.
++    a firefighter is looking down in fire.
```

