# OpenReview forum: "Unsupervised Discovery of Interpretable Latent Manipulations in Language VAEs"
_ICLR.cc/2021/Conference — Reject_

### Official Review · AnonReviewer4 · 2020-10-16
**Simple method but unclear results**

**Rating:** 3
**Confidence:** 4

**Review:**

-------------------
Summary
-------------------
This paper proposes a simple approach to discover interpretable latent manipulations in trained text VAEs. The method essentially involves performing PCA on the latent representations to find directions that maximize variance. The authors argue that this results in more interpretable directions. The method is applied on top of a VAE model (OPTIMUS), and the authors argue that different directions discovered by PCA correspond to interpretable concepts.

-------------------
Strengths
-------------------
- The method is simple, and can be applied on top of existing text VAEs.
- Learning interpretable and controllable generative models of text is an important research area, and this paper contributes to this important field.

-------------------
Weaknesses
-------------------
- There are only mostly qualitative results presented. While I agree that performing quantitative results is difficult with this style of work, the authors could have (for example) adopted methods from the style transfer literature to show quantitative results. These metrics include perplexity (to see how fluent the generations are), reverse perplexity, and style transfer accuracy (this may not be applicable since there is no ground truth "style" in this work, but the ground truth style could be heuristically defined for some transformations, e.g. for singular/plural transformations).
- Human evaluation seems nonideal since it is only tested on 12 people.
- The generations are actually not so good in my opinion? E.g. many of the generations in the appendix are ungrammatical and/or semantically nonsensical.  Again, metrics such as perplexity could quantify the fluency of generated text.
- The method is only applied to one text VAE mode which specifically uses BERT/GPT-2 , so it is not clear if this will generalize to other models (e.g. models trained from scratch).

-------------------
Questions/Comments
-------------------
- In Figure 2, are these the top 4 principal directions? If not, how were these directions discovered?
- "It is known that variational autoencoders trained with a schedule for the KL weight parameter (equation 1) obtain disentangled representations (Higgins et al., 2016; Sikka et al., 2019; John et al., 2019). Since OPTIMUS is also trained with KL annealing, canonical coordinates in its latent space are likely to be disentangled." I believe this is only valid for beta > 1 so it is not really applicable here.
-----------------------
Edit after rebuttal: Thank you for the rebuttal and clarifying some of my questions. I have decided to keep the original score.

---

> ### Author Response · Authors · 2020-11-25
> **Response to Reviewer 4**
>
> Thank you for a detailed review! We address your concerns below:
>
> Weaknesses:
> 1. Thank you for this suggestion! We will look into the evaluation of style transfer models and add more quantitative results in the next revision.
> 2. To our knowledge, having few annotators for evaluation of text attribute manipulation (style transfer in particular) is quite common. For example, in [1] and [2] there are 6 and 10 annotators respectively.
> 3. The generation artifacts are caused not only by the latent shifts, but also by the model itself. We will evaluate this in more detail in the next revision of the paper.
> 4. We also evaluated the method on two models trained from scratch on smaller datasets, namely we trained CP-VAE [3] and the model from [4] on Yelp and Amazon datasets respectively. While some directions are also identifiable (e.g., sentence length and sentiment in case of CP-VAE), the overall generation quality is significantly lower. This decline in fluency is expected: the highest-quality generative models for language have millions of parameters and are trained on massive datasets. To our knowledge, OPTIMUS is the only high-capacity language VAE trained on a large dataset with openly available weights, which is why we mostly evaluate our method on this model.
>
> Questions:
> 1. These are examples of directions obtained with the SNLI model that were manually chosen from a set of generated sentences to highlight the differences between word categories.
> 2. Thank you for the correction! We replaced this motivation with a description of the fully factorized prior distribution of standard Gaussian prior VAEs.
>
> [1] Disentangled Representation Learning for Non-Parallel Text Style Transfer. Vineet John, Lili Mou, Hareesh Bahuleyan, Olga Vechtomova. ACL 2019
>
> [2] Improving Disentangled Text Representation Learning with Information-Theoretic Guidance. Pengyu Cheng, Martin Renqiang Min, Dinghan Shen, Christopher Malon, Yizhe Zhang, Yitong Li, Lawrence Carin. ACL 2020
>
> [3] On Variational Learning of Controllable Representations for Text without Supervision. Peng Xu, Jackie Chi Kit Cheung, Yanshuai Cao. ICML 2020
>
> [4] Controllable Unsupervised Text Attribute Transfer via Editing Entangled Latent Representation. Ke Wang, Hang Hua, Xiaojun Wan. NeurIPS 2019.

---

### Official Review · AnonReviewer2 · 2020-10-27
**Unsupervised Discovery of Interpretable Latent Manipulations in Language VAEs**

**Rating:** 3
**Confidence:** 4

**Review:**

This paper presents a PCA-based latent variable language model for unsupervised latent variable interpretation.

Pros:
1. The authors propose to use PCA to extract the principal components of the results and claim them to be interpretable latent variables.

Cons:
1. The novelty is quite limited. Applying an existing well-known technique to obtain interpretable latent variables is not advancing this domain in the right direction.
2. The explanation of latent variable in this paper is self-justified. The self-defined baselines cannot be convincingly conveyed that latent variable are interpreted. And the baselines are quite weak.
3. In the quality evaluation, the authors do not show how clearly to modify the discovered latent variable to alter the sentences.

Question:
1. How do you encode a sentence in a two-dimensional space? Are both dimension probability?
2. Other than the current quantitative and qualitative analysis, do you think any other quantitative evaluation will be helpful?

---

> ### Author Response · Authors · 2020-11-25
> **Response to Reviewer 2**
>
> Thank you for a thorough evaluation of our paper and constructive feedback! We address your concerns below:
>
> Cons:
> 1. Regarding the novelty: to the best of our knowledge, previous works on unsupervised discovery have not attempted to reveal interpretable latent directions in generative models for language. We believe this is a challenging yet important task that will bring forth more applications as the field develops, similarly to what we observe in image GANs. In addition, our method exploits the availability of the encoder network in VAEs and works directly in the model's latent space. Previous works were applied only to image GANs and required sampling from the latent distribution (which is not necessary for encoder-decoder models) or backpropagation through generated samples (which is not possible with discrete outputs).
> 2. Regarding the baselines: we agree that they are not as strong as one would prefer. However, the choice of baselines here is restricted: the task of unsupervised latent discovery in text generation models has not yet been approached, so the field itself is not quite established. If you have any suggestions on additional methods that would fit the setting of the paper, we would be happy to evaluate them.
> 3. Regarding the modification procedure description: we have updated the text to highlight that each interpretable direction is a vector in the latent space. As a result, applying the shift corresponds to adding this vector to the encoder output.
>
> Questions:
> 1. We believe you are referring to Figure 1: it was meant to give an intuitive explanation of our method; both dimensions correspond to coordinates in an example two-dimensional latent space. The actual method works with 768-dimensional representations of sentences, which are much harder to visualize.
> 2. As suggested by Reviewers 3 and 4, it is possible to measure the fluency of generated outputs (in terms of perplexity) and attribute change quality (in terms of heuristic metrics when we can express the manipulation in simple words).

---

### Official Review · AnonReviewer1 · 2020-10-30
**An interesting examination and exploration of the OPTIMUS VAE model**

**Rating:** 5
**Confidence:** 3

**Review:**

The author propose to use PCA-like method on latent space of VAE models to unsupervisedly detect interpretable direction. The idea is reasonable and practically useful for large-scale pretrained VAE model, i.e. OPTIMUS. This paper has a clear idea and a thorough discussion with related works.

I have some concerns about the model. The proposed model seems requiring a large-scale pretrained model. If the VAE model is just trained on SNIL level, is method still valid?  From the PCA side, it does not require a Gaussian space. So why specifically targeting on VAE model, not just AE model is another confusion. Since the direction is computed based on training data, I kind of feeling of no need of using VAE model.

---

> ### Author Response · Authors · 2020-11-25
> **Response to Reviewer 1**
>
> Thank you for your review! Please allow us to address your concerns below:
>
> 1. In our preliminary experiments, we trained CP-VAE [1] and the model from [2] on Yelp and Amazon datasets respectively. While some directions are also identifiable (e.g., sentence length and sentiment in case of CP-VAE), the overall generation quality is significantly lower. This decline in fluency is expected: the highest-quality generative models for language have millions of parameters and are trained on massive datasets. To our knowledge, OPTIMUS is the only high-capacity language VAE trained on a large dataset with openly available weights, which is why we mostly evaluate our method on this model.
>
> 2. Regarding the applicability of the method to AE models instead of just VAEs. Indeed, one can apply the technique to AEs as well; in fact, one of the models we evaluated was a regular autoencoder (SNLI, $\beta=0$ in Table 1). However, a regular autoencoder is not a proper generative model because its sampling process is not well-defined. Hence, we focus only on variational autoencoders in our work.
>
> [1] On Variational Learning of Controllable Representations for Text without Supervision. Peng Xu, Jackie Chi Kit Cheung, Yanshuai Cao. ICML 2020
>
> [2] Controllable Unsupervised Text Attribute Transfer via Editing Entangled Latent Representation. Ke Wang, Hang Hua, Xiaojun Wan. NeurIPS 2019.

---

### Official Review · AnonReviewer3 · 2020-11-02
**Straightforward idea, hasty  experiments**

**Rating:** 4
**Confidence:** 4

**Review:**

This paper studies latent manipulations in text autoencoders. The authors propose that compared to random and coordinate directions, moving in the PCA directions of encodings of training examples will produce more interpretable text manipulations.

As the idea is straightforward, I'd like to see more in-depth analysis and more solid evaluations. The authors characterize the effects of PCA directions into four types (length, word change, word insertion, and structure enforcement), but for each type only one example is provided. What are the changed/inserted words and what are the enforced structures? Can you give a comprehensive list of them? When are these latent directions applicable and when are they not? For sentences that are not applicable, what effects will they bring?

The only evaluation in the paper is human evaluation of whether a latent direction shift produces interpretable generations. It's conducted on 20 sentences, which is too small to draw any conclusions. The results on the Wikipedia dataset are very poor. You may test the success rate of manipulations in a specific direction (such as word insertion) through automatic evaluation. This can also reveal which manipulations are easier to implement and which are more difficult.

I think with these changes, the paper will be more substantial, instead of spending 4 out of 8 pages on the background like in the current submission. Also, it's more suitable for NLP conferences than ICLR.

---

> ### Author Response · Authors · 2020-11-25
> **Response to Reviewer 3**
>
> Thank you for your review! We address your points below.
>
> 1. For a list of directions, we have added several examples of latent manipulations that were discovered by our method on the SNLI model ($\beta=1$) and separated them into several categories to help the reader understand the differences between groups. These are available in the new section of the appendix.
>
> 2. Regarding the sentences with non-applicable directions: such directions still change the sentences, although the content changes are not as drastic. The degree of content change depends on the magnitude of a shift and is a property of the model: if a particular VAE has latent shifts that can be used only for a subset of texts, any method can reveal them.
>
> 3. On human and automatic evaluation: first, although we use 20 sentences for manipulation, we apply 20 manipulations to each sentence, which gives us 400 initial examples. We reuse these 20 sentences across different manipulations to observe how different latent shifts affect the same sentence. After filtering unchanged sentences, we show 5 examples of each manipulation, which corresponds to 100 transformation examples; for each shift, we have 5 degrees of intensity.
> Second, we agree that testing the interpretability of each manipulation with automatic metrics would strengthen the results. We will measure the fluency of generated sentences and the success of simple transformations in the next revision of the paper.

---

### Decision · Program_Chairs · 2021-01-07
**Final Decision**

**Decision:**

Reject

**Comment:**

This paper proposes a simple method to discover latent manipulations in trained text VAEs. Compared to random and coordinate directions, the authors found that by performing PCA on the latent code to find directions that maximize variance, more interpretable text manipulations can be achieved.

This paper receives 4 reject recommendations with an average score of 3.75. The reviewers have raised many concerns regarding the paper. (i) The idea is straightforward with limited novelty. (ii) There are only mostly qualitative results presented. More in-depth analysis and more solid evaluations are needed. (iii) Human evaluation is too small to draw any reliable conclusion. (iv) The proposed method is only tested on one text VAE, how well it can be generalized to other models remains unclear.

The rebuttal unfortunately did not address the reviewers' main concerns. Therefore, the AC regrets that the paper cannot be recommended for acceptance at this time. The authors are encouraged to consider the reviewers' comments when revising the paper for submission elsewhere.